# *Cryptococcus neoformans* VNII as the Main Cause of Cryptococcosis in Domestic Cats from Rio de Janeiro, Brazil

**DOI:** 10.3390/jof7110980

**Published:** 2021-11-18

**Authors:** Rosani Santos Reis, Isabel Cristina Fábregas Bonna, Isabela Maria da Silva Antonio, Sandro Antonio Pereira, Carlos Roberto Sobrinho do Nascimento, Fausto Klabund Ferraris, Fábio Brito-Santos, Isabella Dib Ferreira Gremião, Luciana Trilles

**Affiliations:** 1Mycology Laboratory, Evandro Chagas National Institute of Infectious Diseases (INI), Oswaldo Cruz Foundation (Fiocruz), Rio de Janeiro 21040-900, Brazil; rosani.reis@ini.fiocruz.br (R.S.R.); isabel.bonna@fiocruz.br (I.C.F.B.); binhofarm@gmail.com (F.B.-S.); 2Laboratory of Clinical Research on Dermatozoonoses in Domestic Animals (Lapclin-Dermzoo)/INI/Fiocruz, Rio de Janeiro 21040-900, Brazil; isabela.maria@ini.fiocruz.br (I.M.d.S.A.); sandro.pereira@ini.fiocruz.br (S.A.P.); isabella.dib@ini.fiocruz.br (I.D.F.G.); 3Mycology Laboratory/National Institute Quality Control in Health (INCQS)/Fiocruz, Rio de Janeiro 21040-900, Brazil; carlos.nascimento@fiocruz.br; 4Pharmacology Laboratory/INCQS/Fiocruz. Av. Brasil, 4365, Manguinhos, Rio de Janeiro 21040-900, Brazil; fausto.ferraris@yahoo.com.br

**Keywords:** cryptococcosis, domestic cats, MLST, molecular epidemiology

## Abstract

Cryptococcosis is a systemic fungal disease acquired from contaminated environments with propagules of the basidiomycetous yeasts of the *Cryptococcus neoformans* and *C. gattii* species complexes. The *C. neoformans* species complex classically comprises four major molecular types (VNI, VNII, VNIII, and VNIV), and the *C. gattii* species complex comprises another four (VGI, VGII, VGIII, and VGIV) and the newly identified molecular type VGV. These major molecular types differ in their epidemiological and ecological features, clinical presentations, and therapeutic outcomes. Generally, the most common isolated types are VNI, VGI, and VGII. The epidemiological profile of cryptococcosis in domestic cats is poorly studied and cats can be the sentinels for human infections. Therefore, the present study aimed to determine the molecular characterization of *Cryptococcus* spp. isolated from domestic cats and their dwellings in the metropolitan area of Rio de Janeiro, Brazil. A total of 36 *Cryptococcus* spp. strains, both clinical and environmental, from 19 cats were subtyped using multilocus sequence typing (MLST). The ploidy was identified using flow cytometry and the mating type was determined through amplification with specific pheromone primers. All strains were mating type alpha and 6/36 were diploid (all VNII). Most isolates (63.88%) were identified as VNII, a rare molecular type, leading to the consideration that this genotype is more likely related to skin lesions, since there was a high percentage (68.75%) of cats with skin lesions, which is also considered rare. Further studies regarding the molecular epidemiology of cryptococcosis in felines are still needed to clarify the reason for the large proportion of the rare molecular type VNII causing infections in cats.

## 1. Introduction

Cryptococcosis is a systemic mycosis that affects a wide variety of mammalian hosts [1]. Most human cases are opportunistic, with risk factors including infection by the acquired immunodeficiency virus (HIV), organ transplantation, lupus, cancer, diabetes, and alcoholism [2,3,4]. These cases are usually caused by members of the *Cryptococcus neoformans* species complex, which is a cosmopolitan agent. Species of the *C. gattii* species complex mainly affect immunocompetent patients and are mostly restricted to tropical and subtropical climates, although in the late 1990s, they expanded to areas with a temperate climate [5,6]. The *C. neoformans*/*C. gattii* complexes classically comprise eight major molecular types: *C. neoformans* var*. grubi*i—VNI and VNII; *C. neoformans* var. *neoformans*—VNIV and the hybrid AD–VNIII; *C. gattii*—VGI, VGII, VGIII, and VGIV [7]. A recent proposal named seven species of the cryptococcosis agents, excluding diploid/aneuploid hybrids [8,9]. Although such monophyletic clades (molecular types) are potentially different species, the proposed nomenclature is still premature, since, as further epidemiological studies have been carried out, new molecular types have been identified, such as VNB [10] of *C. neoformans* and VGV of *C. gattii* [11]. For this reason, the cryptococcosis agents were treated as molecular types in the present study.

These major molecular types differ in their epidemiological and ecological features, clinical presentations, and therapeutic outcomes [12]. The global distribution of the molecular types of the *C. neoformans*/*C. gattii* complexes shows a predominance of VNI in North America (41%), Central America (78%), South America (64%), Africa (72%), Europe (45%), and Asia (77%) [13]. The molecular type VNII also has a worldwide dispersion, but with very low rates (0.8–11%) compared to VNI (27–81%) [14]. The molecular types VGI and VGII are the most frequent molecular types of *C. gattii*. Among all genetic lineages of the cryptococcosis agents, VGI is the most common in Oceania (43%) and the second most isolated in Asia (15%). The molecular type VGII is the second most common in the Americas, with 26% in North America, 21% in South America, and 11% in Central America [13].

The *C. neoformans* species complex is widely dispersed in the environment, especially in soil enriched with the excretion of pigeons or gregarious birds, while the environmental isolation of the members of the *C. gattii* complex has been related to several species of trees. Nevertheless, the isolation of these species from the dwellings of humans has been reported [15,16], emphasizing the possibility of indoor infections.

There is no opportunistic characteristic reported in animal cryptococcosis. In domestic cats, infections with the feline immunodeficiency virus (FIV) and the feline leukemia virus (FELV) are not predisposing factors for the occurrence of cryptococcosis [17]. The infection in cats usually occurs through the inhalation of fungal propagules (rarely from trauma), but the clinical presentation differs greatly from the disease in humans. The most affected area is the nasal cavity with signs of upper respiratory tract infection being common in cats. Dissemination can occur through contiguity to the underlying tissues or via the hematogenous route to any organ, including the skin [18]. However, infection of the central nervous system (CNS) is less frequent than in canines [19] or humans [20].

The epidemiological profile of cryptococcosis in domestic cats is poorly studied. Epidemiological studies of cryptococcosis often include isolates from human hosts and the environment, but rarely isolates from animals. In general, animal cryptococcosis is under-reported. Most of the available literature consists of case reports with some atypical clinical presentations describing the clinical signs, diagnosis, and treatment, but rarely identifying the molecular type, or even the species complex. Additionally, very few animal cases of cryptococcosis have been documented in Latin America [21,22,23]. The inclusion of animal isolates in epidemiological studies provides more complete data on the population circulating in a given geographic area. Therefore, the present study aims to determine the molecular types and subtypes of *Cryptococcus* spp. from domestic cats in the metropolitan area of Rio de Janeiro, Brazil.

## 2. Methods

All viable *C. neoformans* and *C. gattii* strains recovered from cats assisted at the Laboratory of Clinical Research on Dermatozoonoses in Domestic Animals (Lapclin-Dermzoo), Fiocruz, Rio de Janeiro, Brazil, from 2000 to 2019, were included in this study. The owners were asked to investigate the possible environmental sources of infection at their residence and surroundings for the cases that occurred between 2016 and 2019. Clinical and epidemiological data from cats were obtained from the medical records at Lapclin-Dermzoo.

The environmental samples were processed according to the protocol described by Passoni et al. [16]. Briefly, 1 g of each sample was suspended in 50 mL of 0.9% sterile saline with chloramphenicol (400 mg/L), and the supernatant was plated onto a Niger seed agar (NSA). medium The plates were incubated at 25 °C and observed daily for up to five days. Positive phenol-oxidase colonies (indicated by their brown color) were purified and identified using VITEK 2 bioMérieux Systems (VITEK 2, ICB, bioMérieux, Durham, NC, USA). The species complexes *C. neoformans* and *C. gattii* were differentiated on a canavanine-glycine-bromothymol blue medium (CGB).

The ISHAM consensus multilocus sequence typing (MLST) scheme for *C. neoformans* and *C. gattii* was applied [24] to identify the molecular subtypes. Six housekeeping genes (*CAP59*, *GPD1*, *LAC1*, *PLB1*, *SOD1*, and *URA5*) and the IGS1 region were amplified (Biosystems SimpliAmp Thermal Cycle, Applied Biosystems, Singapore) using the PCR conditions previously described [24]. Sequencing was carried out on a Fiocruz Technological Platform Network ABI Sequencer 3730xL. The sequences were manually edited using Sequencher 4.9 software (Gene Codes Corporation, Ann Arbor, MI, USA) and aligned using MEGA 6.06 (http://www.megasoftware.net (accessed on 1 January 2021)). The allele types (ATs) and the sequence types (STs) were identified via sequence comparisons with the *C. neoformans*/*C. gattii* MLST database at http://mlst.mycologylab.org/ (accessed on 1 January 2021). The sequences of all the newly identified allele types were submitted to the *C. neoformans* MLST database. The haplotype and nucleotide diversity were calculated using DnaSP v5.10 (http://www.ub.edu/dnasp/ (accessed on 1 January 2021)).

The mating type was determined by PCR using a couple of mating type-specific (Mat) oligonucleotide primers: Mat-α (forward, 5′-CTTCACTGCCATCTTCACCA-3′ and reverse, 5′-GACACAAAGGGTCATGCCA-3′) and Mat-a (forward, 5′-CGCCTTCACTGCTACCTTCT-3′ and reverse, 5′-AACGCAAGAGTAAGTCGGGC-3′), which generated 101 bp and 117 bp amplification products, respectively. Amplification was performed in a final volume of 25 µL according to Chaturvedi et al. [25]. Standard strains CFRVS 40123 (ATCC 28957) and CFRVS 40142 (ATCC 28958) were used as positive controls.

Identification of the major molecular types was performed by *URA5*-RFLP analysis using the enzymes *Hha*I and *Sau*96I to verify the molecular type of the isolate with dubious results on the MLST [7]. *URA5*-RFLP patterns were assigned visually by comparison with the banding patterns obtained from the standard strains WM 148 (VNI), WM 626 (VNII), WM 628 (VNIII), WM 629 (VNIV), WM 179 (VGI), WM 178 (VGII), WM 175 (VGIII), and WM 779 (VGIV).

Flow cytometry analysis was performed to identify the ploidy of isolates according to Sia et al. [26]. DNA ploidy was accessed using the flow cytometer BD *FACSCalibur*™, and CFRVS 70297 (WM 628) and CFRVS 70302 (WM 178) were used as diploid and haploid standard strains, respectively. The wavelength of the laser beam was 488 nm and 10,000 cells were counted and their fluorescence intensities were measured. The data are shown as histograms, where the abscissa represents the channel numbers in the proportion of their fluorescence intensities and the ordinate shows the number of cells.

## 3. Results

During the study period (2000–2019), 26 cases of feline cryptococcosis and 4 cases of colonization were diagnosed at Lapclin-Dermzoo/INI/Fiocruz. Of these, 16 cases of cryptococcosis and 3 cases of colonization in the oral cavity had the isolates preserved and were included in the study. All 19 studied cases were from the metropolitan area of Rio de Janeiro, Brazil, and most of the cats were mixed breeds (n = 17). More than one strain was isolated from six cats, from different anatomical sites and/or from serial samples. For the other 10 cases, one isolate per cat was selected. The clinical samples included were fragments of skin lesions (3), oral secretions (3), nasal secretions (4), exudates of skin lesions (10), fragments of lymph nodes (2), an exudate from a conjunctival lesion (1), exudates of nasal lesions (3), a fragment of the lung (1), and fragments of the liver (1) (Table 1).

Of the 16 symptomatic cases, 11 had skin lesions (68.75%), i.e., with the lesions being located not near to any mucosa, and 8 had mucosal lesions. The most affected anatomical site was the nasal region (n = 9), followed by the limbs (n = 8). Lesions in other locations were observed to a lesser extent in the face, ear, neck, chest, lips, back, head, abdomen, tail, conjunctiva, and periocular region.

Three residences were visited for environmental sampling. A total of 22 environmental samples were collected from household dust (n = 14), wood (n = 7), and cage dirt (n = 1). *Cryptococcus* spp. were isolated from two residences (cats 18 and 19; Table 1). 

Thirty-six isolates, including twenty-eight clinical and eight environmental isolates, were analyzed. The CGB medium discriminated 29 *C. neoformans* and 7 *C. gattii* strains. After identification, all isolates were deposited in the Collection of Pathogenic Fungi (CFP)/INI/Fiocruz.

Among the 28 isolates of veterinarian origin, 18 were *C. neoformans* VNII, 6 VNI, and 4 *C. gattii* VGII. From environmental origin, five isolates were VNII and three VGII. The MLST analysis identified 17 sequence types (STs) amongst the 35 strains studied and 6 of these had not yet been described: ST616, ST618, ST619, ST620, ST621, and ST639 (Table 1). The CFP560 (VNII) strain showed double peaks in four out of seven studied loci (*CAP59*, *GPD1*, *LAC1*, and *PLB1*), which led to the fact that no ST could be assigned. Flow cytometry analysis confirmed the diploidy status of the strain (Figure 1). The molecular type was determined by *URA5*-RFLP analysis, and the banding pattern produced was compatible with the VNII molecular type. Flow cytometry also identified another five diploid strains (CFP563, CFP564, CFP565, CFP566, and CFP997) corresponding to the molecular type VNII (Table 1).

The analysis of genetic diversity demonstrated clonal populations for the VNI as well as the VNII strains (Figure 2). DNAsp software identified nine haplotypes among the 22 VNII strains. The haplotypic diversity of this group of samples was high (0.892). However, it showed a low nucleotide diversity (0.00117). The VNI genotype, similarly, showed high haplotypic diversity (0.933) and low nucleotide diversity (0.00434), with five haplotypes found in six samples. No recombination was detected in the analysis of the seven loci within these two groups. From the seven VGII strains, five strains belonged to the same ST, resulting in a low haplotype and nucleotide diversity (Hd: 0.524; Pi: 0.00165).

Different STs were identified in four out of five cats that had more than one isolate studied (Table 1). Cat 6 had two isolates identified as ST43 and ST40, and cat 11 had isolates identified as ST618 and ST40. Four isolates were obtained from cat 13: two from skin lesions, ST620, and two from necropsy, ST620 and ST334. Cat 14 had two isolates: from a nasal secretion (ST23) and from a ganglion aspirate (ST621) (Table 1). 

*Cryptococcus* spp. were isolated from two out of the three studied dwellings. The three VGII strains that were isolated from the environmental samples collected in the dwelling of cat 18 all belonged to ST442. The same ST was isolated from the nasal secretion of cat 18 in 2015 after 4 years of discharge, but no lesion was found in the clinical examination. From the decomposed wood collected in the dwelling of cat 19, five VNII strains were isolated and two STs were identified. The two environmental STs and the clinical ST from cat 19 were not identical but were very similar phylogenetically (Figure 2), differing in a single nucleotide in one or two alleles. Determination of the mating types revealed that all isolates were Mat-α. 

## 4. Discussion

Species of the genus *Cryptococcus* have already been isolated from the oropharynx of some birds or the oral cavity of dogs [27,28]. In the present study, we identified three cases of colonization of the oral cavity of cats by isolates of the genotypes VNI, VNII, and VGII. The species complexes *C. neoformans*/*C. gattii* are widely dispersed in the environment and can be isolated from the soil, especially from places with the presence of dry bird excreta (pigeons, parrots, canaries, etc.) and decomposing wood. Colonization of the nasal cavity by *C. neoformans* in cats has been previously reported [27,29], but the colonization of the cats’ oral cavity is demonstrated herein for first time. The cats’ habit of sharpening their claws on tree trunks or licking their hair and paws may be the reason for the presence of the fungus in their oral cavity. The hunting and ingestion of small insects, such as ants, beetles, or honeybees, may also justify the colonization of the oral cavity [30,31,32]. Duncan et al. [29] observed 42% of animals without clinical signs positive for *Cryptococcus* spp. in a serology test (latex particle agglutination) with a negative nasal swab, suggesting that the colonization of the oral cavity may be another gateway to the disease in animals, in addition to the nasal cavity. Furthermore, there are reports of unique gastrointestinal lesions by *Cryptococcus* spp. in dogs and cats [33,34,35], and the absence of a previous signal of infection in the respiratory tract or any other area suggests that the oral route is the entrance door of the fungus.

The most frequent infection site in cats is the nasal cavity, which can occur in 40% of the cases [36]. The occurrence of a single skin lesion may be due to the direct inoculation of fungal propagules on the skin tissue, but its occurrence is atypical [37]. Only one case could be considered of direct inoculation (cat 1). The other cases involved multiple infection sites, as a probable result of hematogenous dissemination [17]. In our study, there was a high percentage (68.75%) of cats with skin lesions, which is considered rare [17]. This can be explained by the fact that the cryptococcosis cases were diagnosed during the feline sporotrichosis epizootic in Rio de Janeiro, that started in 1998, and many of those cases were referred to the Lapclin-Dermzoo/INI/Fiocruz with suspected sporotrichosis due to the skin lesions. This is a limitation in our study that can constitute a selection bias.

The most common molecular type globally is VNI, considering both clinical and environmental isolates [38]. The VNII molecular type has been isolated from all continents, but with a low frequency [14]; it rarely reaches percentages of 24.3%, as recently described in Peru [38]. The present study identified most cat isolates as the VNII molecular type (64.28%). Considering we had a high percentage of cats with skin lesions, the VNII lineage possibly has a greater tendency to cause those skin lesions. Conversely, the high rate of the molecular type VNII identified in our study led us also to consider the possibility of a greater susceptibility of cats to this molecular type. The largest sample of VNII recorded was in a study carried out in Australia, where 45% of the strains of *C. neoformans* isolated from cats were VNII [39]. The difference in susceptibility between hosts (dogs and cats) to the different lineages of *Cryptococcus* spp. was observed in a study where cats were more affected by the molecular type VGIII and dogs by VNI [36]. Another theory would be a greater exposure of cats to the VNII isolates in Rio de Janeiro. The natural habitat of VNI is associated with the excrement of pigeons and decaying trees. However, environmental sources of VNII are not well-defined since most of the known isolates have a clinical origin [40]. Thus, we suggest investigating environmental sources based on the observation of feline habits.

Restricting access to the street did not exempt cats from exposure to the species of the *C. neoformans/C. gattii* species complexes. Animal exposure may be related to internal or external events at the residence [18]. In the present study, the exposure of some cats to the yeasts could be justified by the owner’s report that pigeons would have access to the balcony of the apartment. There was also a report of the use of beach sand in the litter box (cat 8), which could constitute potentially contaminated material contributing to the infection of the cat. The investigation of environmental sources based on the cases of cryptococcosis in cats is important, as it can lead to an intervention to minimize the risks of human and pet infections. The environmental isolates included in the study are related to the cases of feline cryptococcosis. The same sequence type was isolated from cat 18 and its residence, suggesting an indoor acquisition of the infection. On the other site, two STs phylogenetically close were isolated from cat 19 and its residence. 

The species of the *C. neoformans*/*C. gattii* species complexes can share the same ecological niche [41]. In view of this statement, an event of soil disturbance or logging may result in the dispersion of more than one species and/or genotype; eventually, these infectious fungal particles can be inhaled together. Infections with more than one genotype of *Cryptococcus* spp. have already been reported in human patients [42,43]. Our study reports, for the first time, the isolation of more than one sequence type in domestic cats, which were obtained from different infection sites, suggesting that the inhaled strains with different molecular types may have spread to different anatomical sites. 

A single nucleotide polymorphism (SNP) was detected among the isolates from one of the cats (cat 6). The occurrence of microevolution in vitro was reported as a result of successive subcultures for long periods [44] or in vivo during prolonged infections [45]. In vivo microevolution is a hypothesis to be considered, even though the strains of the majority of the cats were obtained before treatment. Cats 11, 13, and 14 have more polymorphic isolates, differing in 6 to 18 nucleotides (including up to two loci), a finding that emphasizes the hypothesis of inhalation of more than one genotype. 

The low molecular diversity supports the clonal population of *C. neoformans* VNI and VNII, and the absence of the mating type **a** serves as additional evidence of the lack of recombination among the isolates. However, in the present study, six diploid strains, molecular type VNII MATα, were identified. Diploidy is usually transient and precedes several types of reproduction, such as sexual reproduction and same-sex mating type (endoreplication and hybridization) [46]. Nonetheless, the diploid condition may remain relatively stable, for unknown causes, or there may be a loss of chromosomes (aneuploidy). 

Despite the population clonality, the isolates demonstrated a high haplotypic diversity, with 17 sequence types being identified among the 35 strains studied. The sequence types ST5, ST23, and ST77 of *C. neoformans* VNI seem to be cosmopolitan and have also been identified in several countries. The ST5 is the dominant genotype in several populations in China [47] and is significantly present in other Asian countries (Japan, South Korea, Vietnam, and Thailand). This sequence type has also been reported from North America, South America, Europe, South Africa, and the Ivory Coast [42,47]. ST23 and ST77 were described in four continents: Asia, Africa, America, and Europe [48,49,50]. The most frequent sequence type identified in the present study was ST43 (VNII), which was isolated from four cats and from one wood sample. It occurs in Germany, Japan, Nigeria, and the Ivory Coast [42,48,51,52,53]. There are very few molecular epidemiology studies of the molecular type VNII because its isolation worldwide is rare. Therefore, the small number of samples makes the statistical analysis of the data impossible; however, this would be needed to clarify the influence of molecular subtypes on clinical evolution and outcome.

ST20 (VGIIa) is considered a highly virulent subtype and it was the main sequence type of the Vancouver Island epidemic in 1999, which later expanded to the northwestern U.S. states, Washington and Oregon [54,55]. ST20 is frequently found on the American continent and considered an endemic in the Brazilian Amazon region [56,57]. In the present study, we reported a feline case of oral colonization, which never left its residence/domicile in the city of Rio de Janeiro, in the southeast region of the country, with the molecular type VGII, which shows the presence of this genotype in this region.

Further studies regarding the molecular epidemiology of cryptococcosis in felines are still needed to clarify the reason for the large proportion of the rare molecular type VNII identified in this study; whether it is due to the preference of VNII isolates to cause skin lesions in cats or a greater susceptibility of cats to be infected by VNII isolates remains unknown.

## Figures and Tables

**Figure 1 jof-07-00980-f001:**
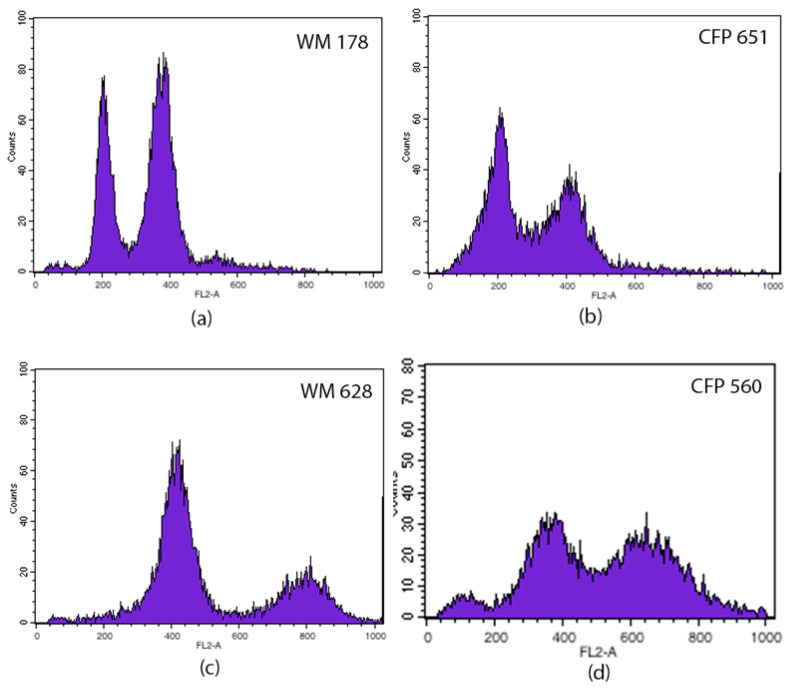
Histograms of DNA content from representative haploid (**b**) and diploid (**d**) strains, and haploid (**a**) and diploid (**c**) standard strains. The abscissa represents the channel numbers in the proportion of their fluorescence intensities and the ordinate shows the number of cells.

**Figure 2 jof-07-00980-f002:**
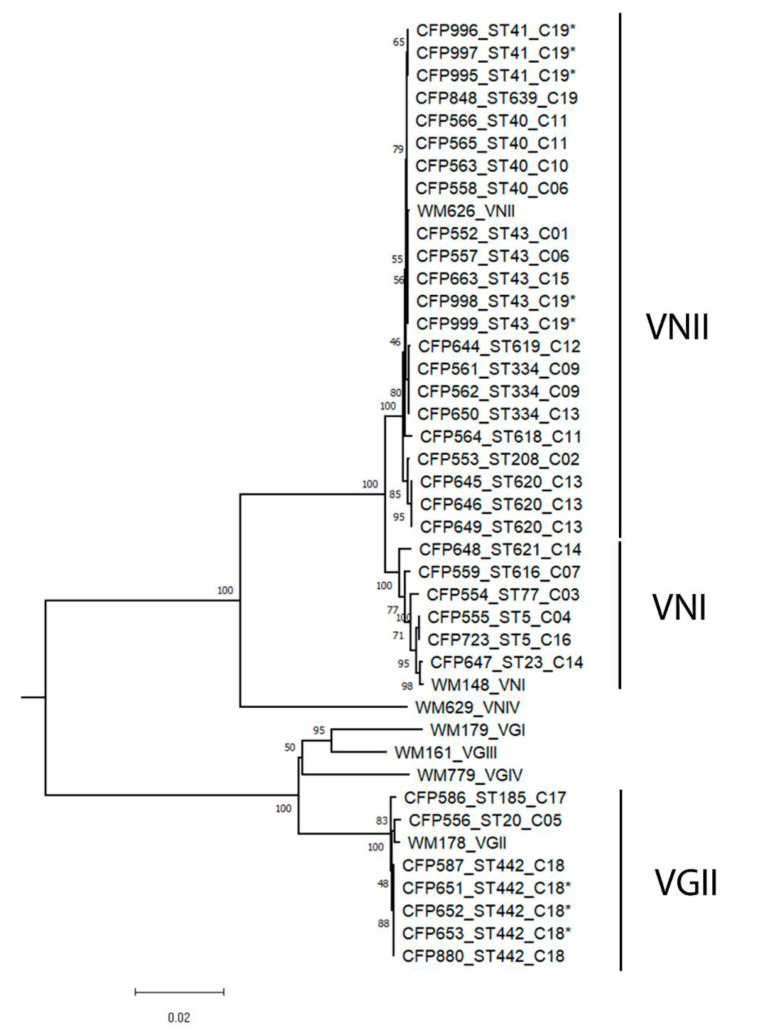
Unrooted neighbor-joining tree inferred from the combined MLST sequences of *CAP59*, *GPD1*, *LAC1*, *SOD1*, *URA5*, and *PLB1* genes and the IGS1 region of the 36 strains investigated in this study, and the 7 standard strains. Numbers above the branches are bootstrap values obtained from 1000 pseudoreplicates. Samples are identified with the Culture Collection number (CFP), followed by the subtype (ST) and the number of the cat (C). (*) Environmental isolates.

**Table 1 jof-07-00980-t001:** Data of *Cryptococcus* spp. isolates from the cats studied according to year and anatomical sites of isolation, molecular type, and subtype.

Felines	Year of Isolation	Strain Number (CFP)	Sample	MT	ST	Ploidy
C1	2000	552	Fragment of skin lesion	VNII	43	Haploid
C2	2002	553	Oral secretion	VNII	208	Haploid
C3	2002	554	Oral secretion	VNI	77	Haploid
C4	2003	555	Fragment of lymph node	VNI	5	Haploid
C5	2002	556	Oral secretion	VGII	20	Haploid
C6	2004	557	Nasal secretion	VNII	43	Haploid
	2004	558	Fragment of skin lesion	VNII	40	Haploid
C7	2004	559	Nasal secretion	VNI	616	Haploid
C8	2004	560	Exudate of skin lesion	VNII		Diploid
C9	2005	561	Exudate of skin lesion	VNII	334	Haploid
	2005	562	Exudate of skin lesion	VNII	334	Haploid
C10	2005	563	Exudate of skin lesion	VNII	40	Diploid
C11	2007	564	Exudate of skin lesion	VNII	618	Diploid
	2007	565	Nasal mucosal lesion	VNII	40	Diploid
	2007	566	Exudate of skin lesion	VNII	40	Diploid
C12	2001	644	Fragment of skin lesion	VNII	619	Haploid
C13	2016	645	Exudate of skin lesion	VNII	620	Haploid
	2016	646	Exudate of skin lesion	VNII	620	Haploid
	2016	649	Fragment of liver	VNII	620	Haploid
	2016	650	Fragment of lung	VNII	334	Haploid
C14	2016	647	Nasal secretion	VNI	23	Haploid
	2016	648	Fragment of lymph node	VNI	621	Haploid
C15	2012	663	Exudate of nasal lesion	VNII	43	Haploid
C16	2001	723	Conjunctival lesion	VNI	5	Haploid
C17	2015	586	Exudate of nasal lesion	VGII	185	Haploid
C18	2015	587	Exudate of skin lesion	VGII	442	Haploid
	2016	651	Dust	VGII	442	Haploid
	2016	652	Dust	VGII	442	Haploid
	2016	653	Dust	VGII	442	Haploid
	2019	880	Nasal secretion	VGII	442	Haploid
C19	2019	848	Exudate of skin lesion	VNII	639	Haploid
	2019	995	Wood	VNII	41	Haploid
	2019	996	Wood	VNII	41	Haploid
	2019	997	Wood	VNII	41	Diploid
	2019	998	Wood	VNII	43	Haploid
	2019	999	Wood	VNII	43	Haploid

MT—molecular type, ST—subtype.

## Data Availability

Sequences of the allele types and their corresponding sequence types are publicly available at The International Fungi MLST Database (https://mlst.mycologylab.org/, accessed at 1 January 2021).

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
