# Peer review of "Cryptococcus neoformans VNII as the Main Cause of Cryptococcosis in Domestic Cats from Rio de Janeiro, Brazil"

_jof, 2021, doi:10.3390/jof7110980_

Round 1
Reviewer 1 Report
Your research is very interesting. It provides new insights into the molecular biology of cryptococcosis in general and of feline cryptococcosis in particular.Author Response
1. Your research is very interesting. It provides new insights into the molecular biology of cryptococcosis in general and of feline cryptococcosis in particular.
R. Thank you so much for the review.
Reviewer 2 Report
Dear Authors:
This is a very interesting article about cryptococcosis in cats.
I have only find very few mistakes:
a) Abstract: line 20: It says VGI, VGII, VGIII and VNIV, please correct.
b) In Methods: there are some units without spaces after the figures (lines 96, 97, 98)
c) Discussion: in the same paragraph you wrote Cote d'Ivoire and then Ivory Coast (please unify).
Title: I consider that 18out of 28 Cryptococcus neoformas VNII is not enough to say that it is the "leading cause of Cryptococcosis in domestic cats".
Author Response
Thank you very much for your corrections and suggestions. Please, verify the responses below:
a)Abstract: line 20: It says VGI, VGII, VGIII and VNIV, please correct.
R. It is corrected.
b) In Methods: there are some units without spaces after the figures (lines 96, 97, 98)
R. They are corrected.
c) Discussion: in the same paragraph you wrote Cote d'Ivoire and then Ivory Coast (please unify).
R. It is corrected.
d) Title: I consider that 18out of 28 Cryptococcus neoformas VNII is not enough to say that it is the "leading cause of Cryptococcosis in domestic cats".
R. We are aware that the number of samples is limited, but we intended to call the attention to the fact that most infections (60%) were caused by the molecular type VNII. Therefore, we replaced the word “leading” with a less intense one, “main cause".
Reviewer 3 Report
The authors investigated genetic structure of Cryptococcus neoformans strains isolated from domestic cats in Brasil. The topic is very interesting because highlights the role of animals as sentinel for human/animal pathogens acquired from the environment.
The study is well written and clear. The Methods well described and results well discussed. The manuscript is suggested to be published.
I have no additional suggestions or comments.
Author Response
Thank you so much for the review.
Reviewer 4 Report
I think this is a very useful report which adds to knowledge.
I have a few issues and suggestions. They will involve a bit of extra work. But I think as a result the manuscript will be enhanced.
- In Table 1 – the list of isolates; my impression is the veterinary data available is LIMITED. Maybe by having a veterinarian on the paper we could pick up some extra detail. The Sample often says, “Exudate of skin lesion” or “Oral secretion” or “Nasal secretion”. These terms do not inspire confidence. A skin lesion might be next to the nasal cavity because there is primary rhinosinusitis. Is it possible to have more convincing clinical detail about the cases? Specifically, i want to know if the skin lesions were in skin close to the nares or bridge of the nose?
- Discussion line 208 – change pawns to paws !!
- line 214-215 – there is a better paper in JSAP by Luke Johnson and colleagues that should be quoted here.
Abdominal cryptococcosis in dogs and cats: 38 cases (2000‐2018)
L Johnston, B Mackay, T King, MB Krockenberger, R Malik, A Tebb
Journal of Small Animal Practice 62 (1), 19-27
- There is a wide range of crypto isolates – VNI VNII and VGII. Could the authors plot all the isolates with a different symbol for each type – and place them on a map of Rio or Brazil. I am interested if there is any spatial clustering of species and biotypes, or their distribution is random.
- It would be good to have susceptibility data for the isolates tabulated somewhere. I am assuming the VNI and VNII will be susceptible to fluconazole amphotericin B and 5 FC but will the VGII isolates show hetero-resistance to fluconazole.
- Were there any coinfection with cryptococcosis and sporotrichosis?
- Were the cats treated and do we know their outcomes?
- Was there IMMY lateral flow or latex agglutination titres for any of the cats?
Author Response
Please see the responses in the attached file.
